# Biomonitoring of Hg^0^, Hg^2+^ and Particulate Hg in a Mining Context Using Tree Barks

**DOI:** 10.3390/ijerph18105191

**Published:** 2021-05-13

**Authors:** Sandra Viso, Sofía Rivera, Alba Martinez-Coronado, José María Esbrí, Marta M. Moreno, Pablo Higueras

**Affiliations:** 1Instituto de Geología Aplicada, Universidad de Castilla-La Mancha, 13400 Ciudad Real, Spain; sandra.limon7.sv@gmail.com (S.V.); Sofia.Riverajurado@uclm.es (S.R.); 2Independent Researcher, C/Madrid 18, 13500 Ciudad Real, Spain; alba162@hotmail.com; 3Escuela Técnica de Ingenieros Agrónomos, Universidad de Castilla-La Mancha, 13071 Ciudad Real, Spain; martamaria.moreno@uclm.es

**Keywords:** particulate-bound mercury, plant uptake, biomonitoring, Almadén, thermal speciation, cinnabar, total atmospheric mercury, leaves, barks

## Abstract

The biomonitoring of atmospheric mercury (Hg) is an important topic in the recent scientific literature given the cost-benefit advantage of obtaining indirect measurements of gaseous Hg using biological tissues. Lichens, mosses, and trees are the most commonly used organisms, with many standardized methods for some of them used across European countries by scientists and pollution regulators. Most of the species used the uptake of gaseous Hg (plant leaves), or a mixture of gaseous and particulate Hg (mosses and lichens), but no method is capable of differentiating between main atmospheric Hg phases (particulate and gaseous), essential in a risk assessment. The purpose of this work was to evaluate different uptake patterns of biological tissues in terms of atmospheric Hg compounds. To accomplish this, the feasibility of two plant tissues from a tree commonly found in urban environments has been evaluated for the biomonitoring of gaseous Hg species in a Hg mining environment. Sampling included leaves and barks from *Platanus hispanica* and particulate matter from the atmosphere of the urban area around Almadén (south-central Spain), while analytical determinations included data for total Hg concentrations in biological and geological samples, Hg speciation data and total gaseous Hg (TGM). The results allowed us to identify the main Hg compounds in leaves and bark tissues and in atmospheric particulate matter, finding that leaves bioaccumulated only gaseous Hg (Hg^0^ and Hg^2+^), preferably during daylight hours, whereas the barks accumulated a combination of TGM and particulate bound Hg (PBM) during the day and at night. Subsequent merging of the atmospheric Hg speciation data obtained from leaves and barks allowed indicative maps of the main sources of TGM and PBM emissions to be obtained, thereby perfectly delimiting the main TGM and PBM sources in the urban area around Almadén. This method complements TGM biomonitoring systems already tested with other urban trees, adding the detection of PBM emission sources and, therefore, biomonitoring all Hg species present in the atmosphere. Scenarios other than mining sites should be evaluated to determine the utility of this method for Hg biospeciation in the atmosphere.

## 1. Introduction

Although the mercury (Hg) cycle involves all environmental compartments, its transport fluxes preferably take place in the atmosphere. Monitoring these mercury fluxes has remained a scientific challenge over the past century given the low concentrations at which Hg is present in the atmosphere, a fact that has required significant technological progress to lower the detection limits of measurement equipment. The first monitoring systems for Hg in air took advantage of the ability of this element to form amalgams with other compounds and metals, such as Au, Ag or Cu, to capture atmospheric Hg in traps, where it could be quantified using classical analytical techniques. For example, Stock and Heller [1] described a method to capture Hg in chlorine water, while Stock and Cucuel [2] proposed to dissolve Hg in chlorine water and to deposit it electrolytically on a copper wire [3]. These approaches allow gaseous Hg data to be obtained for periods of time significant enough to allow an occupational exposure assessment but insufficient to understand flows in terms of trends and quantities. All this trapped Hg was determined by Atomic Absorption Spectrometry in a gaseous state, thus suggesting the possibility of determining it directly using gaseous samplers [4]. Numerous devices (both static and portable) have been developed to measure gaseous Hg directly. Although some of them were able to distinguish gaseous elemental Hg (GEM) from reactive gaseous Hg (RGM) and particulate-bound Hg (PBM), most could only quantify GEM or total gaseous Hg (TGM), which is the sum of GEM and RGM [5]. The data provided by these monitoring devices allowed main Hg flows and deposition rates (both dry and wet) to be identified and quantified, although they were expensive and often used for only brief time-periods, thereby lacking representativeness. As such, the next advance in monitoring systems involved a step back: once again, to evaluate the possibility of capturing Hg in physical or biological traps to obtain reliable and representative Hg data over long periods of time in a simple and inexpensive manner [6]. There are two main approaches to this, namely the use of passive samplers and biomonitoring using biological tissues. The main advantages of using these monitoring systems lie in the greater representativeness of the data obtained, in terms of both time and space. Many tissues have been used for these purposes, mainly mosses and specially lichens, but also plants, as their aerial part, have the ability to absorb Hg directly from the atmosphere. These biomonitoring strategies provide data about GEM and TGM, although the proportion of PBM uptake by biomonitors is usually underestimated. Mosses and lichen uptake a variable proportion of PBM instead of high proportions of GEM [7,8], and some plant tissues can be an excellent candidate for biomonitoring PBM. Amongst these, one promising option involves the use of tree barks due to their low permeability to gases and, subsequently, low GEM exchange capacity, which diminishes the role of GEM as a confounding variable [9]. Although barks are usually impermeable to gases, lenticels provide the possibility to exchange gases via the periderm. These circular, oval, or elongated areas represent an exchange channel for O_2_, CO_2_ and H_2_O (gas). As such, tree barks may be an excellent candidate to biomonitor particulate Hg due to their impermeability to gases and the existence of lenticels.

The present study aims to evaluate tree barks as a biomonitor for PBM in a mining context, an environment with high possibilities of generating particulate matter, which contains high concentrations of PBM of mining origin that is easily traced by its composition. To accomplish this main purpose, bark and leaves from *Platanus hispanica*, a common urban tree in Mediterranean regions, were studied to determine TGM and PBM in Almadén, a very well-known mining locality in the scientific literature. *Platanus hispanica* was chosen as the study specie because it is the most represented spatially in the city of Almadén and also because it is ideal for bark sampling (it can easily be detached from the trunk during the warm months).

## 2. Materials and Methods

To achieve the objectives of the research, it was decided to use the city of Almadén as a study site due to its well-known sources of mercury emissions, as already described in previous publications [10,11,12]. The research design included the acquisition of comparable data for total Hg (THg) in the bark and leaves of *Platanus hispanica* (plane trees), as well as TGM and PBM data from the city of Almadén. To correctly understand these data, micrometeorological data from the Almadén area were also acquired.

Mining activities have gone on at Almadén for more than 2000 years, which has resulted in a huge legacy of Hg dispersion in all environmental compartments. The evolution of GEM levels in the city of Almadén evolved from the highest levels at the end of 20th century, with 100–5000 ng m^−3^ produced by the still active metallurgical activity [12], to lower ones after mine closure, the cessation of metallurgical activities, and restoration of the waste dump (0.8–686.9 ng m^−3^) [10]. There are various sources of mercury emissions in the urban area of Almadén (Figure 1): the mining-metallurgical complex can be considered as the main Hg emission source, since it still has derelict facilities with abundant Hg in a liquid state that generates intense emissions; the most important secondary emission sources are the large dump, which is currently confined under geotextiles and high-density plastics [13], but which continues to generate low emissions [11]; the contaminated soils around it, both to the south and in an old metallurgical area to the north; an abandoned illegal landfill; several dirt roads that have been restored using waste from the dump, which still emit Hg gas; and, finally, several urban monuments containing large blocks of cinnabar. The contaminated soils mainly contain cinnabar, metacinnabar and Hg_2_Cl_2_, and there are high levels of methyl-Hg in the sediments of the Azogado stream [14].

Fourteen sampling sites were used to collect leaf and bark samples, as well as to measure gaseous mercury concentrations and PBM (Figure 1). The leaf and bark samples were composite samples (a mixture of subsamples collected from contiguous trees, in a circular area 15 m in diameter). Leaf samples were collected using a telescopic stainless steel leaf cutter, while bark samples were separated from the trunk using a knife, ensuring to sample the barks at a height of between 1.5 and 2 m. Composite samples always have included barks and leaf pieces of at least two trees, some 15–20 barks pieces, and 40–50 leaves for each sample. A reference site (site 01) was included in Chillón, an agricultural village 2.4 km northwest of Almadén. Preparation of the biological samples began with a thorough washing with deionized water, followed by drying in an oven at 35 °C to avoid Hg losses. The leaves were then crushed with scissors and the bark was crushed using a laboratory crusher. Both types of samples were kept in polypropylene bottles sealed with parafilm until analysis to avoid any exchange of Hg with the laboratory atmosphere.

THg concentrations in the biological samples were determined by atomic absorption spectrometry with Zeeman effect, using an RA-915M instrument (Lumex, St.Petersburg, Russia) equipped with a PYRO-91 pyrolyzer. This technique allows the quantification of THg in biological samples without interference from organic compounds [15] in a wide range of concentrations and with a low detection limit (10 ng g^−1^). Analyses were performed in triplicate and certified reference materials (SRM 1515–Apple leaves) were analyzed to check precision and accuracy, obtaining recovery rates in the range (81.8–83.9%).

The Hg species present in the biological samples were determined using the same technique as for THg and a heating ramp from 35 to 600 °C to distinguish the different Hg compounds by thermal desorption [16]. As this technique presents serious limitations for the comparison of desorption temperatures, including the environmental conditions in which it is carried out, the equipment used, the relative position of the samples in the boat with respect to the temperature measurement thermocouple and the design of the heating ramp and its temperature increase point, it is necessary to analyze pure compounds (Hg^0^, metacinnabar, methyl-Hg, Hg bound to humic acids, cinnabar and schuetteite) under the same analytical conditions as for the leaf and bark samples. Pure compounds were selected based on data provided by synchrotron inorganic speciation [17].

TGM was monitored by atomic absorption spectrometry, using a portable Lumex RA-915M system (Lumex, St.Petersburg, Russia). Each measurement was made after a waiting period of 5 min to allow acclimatization of the equipment and to avoid the existence of air currents produced by the sampling equipment upon arrival. The measurement period was two minutes, at a height of 1.5–2 m in the area of the sampling site. All measurements were performed in less than two hours, between 12:00 and 14:00 CET, taking advantage of the period with the highest micrometeorological stability to ensure the comparability of the data obtained.

Particulate matter was sampled using a high-volume sampler (MCV CAV-A/mb) placed on the roof of a small building at the Almadén Mining School (point G in Figure 1). Samples were accumulated on Whatman QMA quartz filters (diameter 47 mm). Two sampling cycles were used, one of them covering 24 h and the other one covering day or night (15 and 9 h, respectively). Sampling equipment was maintained at a constant flow rate of 30 m^3^ h^−1^ for all samples. Once the sampling time had been completed, filters were dried for 24 h in an oven at 35 °C, then weighed on an analytical balance to obtain total suspended particles by weight differences. The THg concentration was measured directly by pyrolysis in an AMA-254 Atomic Absorption Spectrometer (LECO, Praha, Czech Republic) using half of each filter sample, following the same methodology as in [18].

A geostatistical treatment was applied to obtain the THg distribution maps for biological samples and GEM, using Golden Software Surfer 9 for this purpose. Block kriging was used to interpolate the data and the mapping thresholds were calculated using the average to delimit background values, and the average plus two standard deviations to visualize anomalous values.

A basic pollution index was calculated to discriminate Hg data from leaves and barks. This was expressed as the ratio C_sample_/C_background_, using site 01 as the background site.

Micrometeorological parameters were measured using a fully automatic Davis Vantage Pro station (Davis, Hayward, CA, USA). Parameters measured directly were temperature, relative humidity, wind speed, wind direction, barometric pressure, rain, solar radiation, and UV radiation.

## 3. Results

### 3.1. Hg in Biological Samples and in the Atmosphere

Sample 01 was taken in Chillón (2.4 km northeast), a nearby rural town with no Hg deposits or any type of mining or metallurgical activity, which was therefore considered to be local background. Tree barks in Almadén showed THg concentrations in the range 42.5–1666.0 ng g^−1^ (local background: 23.2 ng g^−1^), whereas tree leaves showed THg levels in the range 90.1–1231.0 ng g^−1^ (local background: 90.2 ng g^−1^; Table 1). Pollution indexes were higher in bark than in leaves (>70 ng g^−1^ in bark samples and a maximum of <14 ng g^−1^ for leaves). These differences suggest differences in Hg uptake capacity between bark and leaves tissues, or differences in the uptake of Hg compounds by each, possibly TGM for leaves and PBM for bark. The second option can be explained by the fact that the residence time for TGM in the atmosphere is significantly greater than that for PBM, as it is also the case for dispersion distances, which result in a wider dispersion for TGM than for PBM around the main emission sources in Almadén (Figure 1). Although Hg data for plane tree bark and leaves are scarce in the scientific literature, some data have been provided for bark from urban plane trees (*Platanus acerifolia*) exposed to traffic pollution [19], with levels of 63–86 ng g^−1^ in European cities and 62–160 ng g^−1^ in Polish cities, thus meaning that 62.4% of samples considered herein had Hg concentrations higher than those from these urban concentrations. These reference values cannot be considered to be background values since the samples concerned were exposed to traffic pollution and, especially, domestic heating systems. However, they can be used for the comparison since the trees in Almadén are also located in urban sites and it is likely that some of the Hg accumulated in the bark comes from traffic or heating systems rather than just mining sources.
ijerph-18-05191-t001_Table 1Table 1Statistical summary of THg concentrations for bark and leaves, expressed in ng g^−1^. Site 01 is considered as background. The pollution index (PI) is expressed as the ratio C_sample_/C_background_. Reference concentration as referred for *Platanus acerifolia*. SD: standard deviation.SiteBark THgSDPILeaf THgSDPIReference00524.077.422.690.142.41.0
0123.25.51.090.222.91.0
02627.6119.127.11231.0141.813.6
03655.670.128.3682.855.57.6
04248.744.310.7604.9177.96.7
05472.3116.220.4985.9196.310.9
061666.0221.171.8497.953.45.5
07756.754.032.6471.847.55.2
08197.020.88.5237.217.92.6
09140.024.96.0136.713.11.5
10145.521.76.3335.553.63.7
11236.731.910.2198.027.22.2
1273.118.13.2129.427.31.4
13108.021.34.7118.111.71.3
1442.52.61.8263.486.62.9
European cities63–86




[19]Poland62–160




[19]

The distribution of the THg values with respect to the main Hg emission sources is displayed in Figure 2. When looking at the THg and TGM data at the same level of detail as the tree data (Figure 2A–C), the influence of the main emission source is perfectly captured by the sampling network, although the same cannot be said for the minor emission sources (Figure 1). The relationship between the data distributions of leaf THg (Figure 2B) and TGM levels using the same sampling sites (Figure 2C) is also clear, especially considering the relative location of maximum values on the main gaseous Hg source. This relationship was expected considering that the primary Hg uptake pathway for leaves is the atmospheric pathway and that the translocation of Hg captured by roots can be considered negligible [20,21]. On the other hand, the distribution of THg in bark (Figure 2A) shows important differences in terms of the maximum content, which was found about 500 m southeast of the main dump. To investigate the possibility that there is a secondary zone with high levels of TGM close to this point, a more detailed monitoring was carried out throughout the urban area of Almadén (Figure 2D). In this detailed map, the area with values below the sampling average was found to be reduced, and new zones of maximum TGM levels were found, thus reflecting both the influence of the main emission source and also of secondary emission sources, especially the cinnabar monument (5 in Figure 1) and the illegal urban waste dump. However, this map does not show a TGM maximum around the maximum THg level of bark, as expected.

Table 2 shows a single Pearson correlation study. Main significative correlations have been found between THg concentrations in barks and TGM in survey display in Figure 2D, and THg concentrations in leaves with both TGM distribution maps based on TGM sites data and a complete TGM survey (Figure 2C,D, respectively). Gaseous Hg concentrations appear as a key factor in THg in leaves, but showing minor importance in barks. The absence of significative correlations of THg in leaves and/or barks with TGM in night hours suggest that gaseous Hg uptake process is linked the opening of stomata.

The PBM data gave an average of 4.85 ng m^−3^ for a complete year, with most of the values being below 8 ng m^−3^ and maxima as outliers in spring and winter (Figure 3). The seasonal evolution showed higher PBM levels in winter, with small differences between autumn, spring and summer. On a daily cycle basis, higher levels were found at night than during daylight hours. To understand these findings, it should be remembered that the sampling equipment was installed on the roof of a small building inside the Almadén mining school, with no nearby soils that could emit particles with Hg. As such, the PBM collected corresponds to particulate matter transported from nearby emission sources: the soils of the pine strip located south of this school and all emission sources located to the west of the mining-metallurgical complex. Micrometeorological parameters show stagnation conditions during the night, in terms of wind speed, thus suggesting that dry deposition of particulate matter is the main process involved in these PBM data. As such, these PBM data can be considered to be sufficiently representative of the PBM present in the Almadén atmosphere, although it must also be noted that local emission sources may produce local differences in this distribution by season and by day-night cycle.

### 3.2. Speciation Data

Thermal desorption of Hg compounds represents a direct speciation method, thus enabling the main Hg compounds to be distinguished and even quantified. Only two Hg compounds have been identified in this study: Hg bound to organic matrix, with a desorption temperature in the range 218–225 °C, and cinnabar, with a desorption temperature in a range 292–300 °C (Figure 4). Leaf samples show a single peak, whereas bark samples and particulate matter show two peaks. The proportions of Hg bound to organic matrix and cinnabar were not the same, being around 30–70% in bark samples and 40–60% in particulate matter, respectively. To ensure the identification of these species, pure Hg compounds were analyzed under the same analytical conditions, thus obtaining desorption temperatures of 188 °C for methyl-Hg from tuna fish (BCR 463), 220 °C for Hg bound to humic acids in a soil matrix (internal reference compound), 221 °C for Hg bound to organic matrix in olive leaves (internal reference compound) and 305 °C for cinnabar (internal reference compound).

## 4. Discussion

This study is based on use of the Hg cycle at the soil-plant-atmosphere interface to evaluate the feasibility of using some plant tissues as biomonitors for PBM. The Hg concentration data considered were the THg concentrations in leaves and bark from *Platanus hispanica*, the Hg concentration in the atmosphere, both gaseous and particulate, as well as Hg speciation data in leaves, bark and particulate matter. Soil data would be the only representative of the Hg available for uptake by the roots of the tree, but it was not considered feasible to add because tree rhizosphere is too deep to be sampled. In addition, translocation of Hg from roots to the aerial part can be considered to be negligible [20].

A single Pearson correlation analysis shows only a significant correlation (r^2^ = 0.89, ρ < 0.01) for THg in leaves *vs* GEM, as expected, given the strong similarities between their distribution in the Almadén urban area (Figure 2B,C). The role of leaf stomata in Hg interexchange [22] suggests that this tissue is a good alternative for biomonitoring TGM emissions around an important emission source. The use of *Platanus hispanica* leaves allowed us to identify isolated emission sources in urban environments (for example, waste-management industries treating Hg-containing residues under conditions that result in inadvertent release into the atmosphere), as well as monitoring the annual evolution of remediation works in contaminated areas, by taking advantage of the deciduous nature of the chosen tree. However, several factors must be taken into account that can limit its use or the interpretation of the data obtained, namely that Hg exchange at the tree-atmosphere interface is bidirectional [21,22]; the relation between exposure and uptake (or re-emission) may not be linear; there is no standardized method, and therefore the results are rarely comparable; the Hg cycle in the study area may be complex and may contain interactions between environmental compartments that do not allow correct interpretation of the data; and the fractionation of atmospheric Hg between the different species may lead to errors in interpretation [23]. Despite this, the application of this biomonitoring method in Almadén has allowed us to locate the main source of Hg gaseous emission in the area and to satisfactorily delimit its area of influence (orange in Figure 2B). On the other hand, it has not proved possible to identify the areas of influence of the minor emission sources detected in the direct measurements using portable spectrometry (Figure 2D), which may be a serious limitation as regards the monitoring of remediation works as it would not be capable of detecting secondary emission sources (temporary accumulation of waste, breakage of isolation covers, etc.). In these studies, it is recommended to perform a basic speciation of the bioaccumulated Hg in leaves to confirm that the THg data correspond only to gaseous Hg and there is no contribution from PBM, to elucidate Hg major sources. This biomonitoring method using native trees has been applied successfully in Almadén [24], around a chloralkali plant in Rumania [25], and in background areas using *Pinus nigra* [26], while other authors have used transplanted *Pinus nigra* specimens with reliable results [27].

The concentrations of THg in bark, however, did not show significant correlations with those of TGM (r^2^ = 0.18, ρ = 0.49), or with the concentrations in leaves (r^2^ = 0.42, ρ = 0.11), thus suggesting the existence of fundamental differences in the Hg uptake process in bark *vs* leaves. The distribution map identifies a secondary source of gaseous Hg emission to the southeast of the main source rather than the main source itself (Figure 2A), This secondary source comprises contaminated soils containing up to 176 mg kg^−1^ of THg [28]. Thermal speciation analyses verify that bark samples contain both the labile Hg compound identified in leaves and cinnabar from the contaminated soils adjacent to sampling site 06 (Table 1), thus confirming the participation of PBM in the process of Hg uptake by bark. Particulate matter samplings at the School of Mines (point 7 in Figure 1) showed significant concentrations of PBM in the Almadén area (0.8–16.9 ng m^−3^), with a fractionation of Hg compounds consistent with that found in bark. *Platanus hispanica* bark can be considered to be fundamentally impermeable to gases thanks to the periderm (phellem–phellogen complex), which acts as a barrier to the transfer of gases and liquids. In this kind of trees, the phellem is interspersed with lenticels, aerenchymatous areas of cork that are considered pathways for the exchange of water vapor and other gases (O_2_, CO_2_) [9]. Most of the bark surface is constituted by phellem, the permeability of which for gases is very much lower than that for water, and whose main gas-exchange pathway are lenticels [29]. As such, the uptake of TGM by bark occurs via the lenticels, pores that are regulated in a seasonal cycle rather than the daily cycle for the stomata of the leaves, for example [30]. This important difference implies differences in the period of uptake of Hg from leaves vs bark—diurnal in the case of leaves, seasonal in the case of bark—which limits its application and, at the same time, offers interesting possibilities for TGM biomonitoring in the night period using bark. Concentrations exceeding the recommended levels for chronic exposure often occur at night, when the wind and the dilution capacity of the atmosphere is reduced [10], thus meaning that tree leaf biomonitors are likely to underestimate the TGM present in the local atmosphere, bioaccumulating Hg only in diurnal hours. In addition, bark uptake both gaseous Hg and PBM, in a proportion of 30% according to our speciation analysis. A study in *Pinus nigra* from the Mount Amiata region, which also hosts important Hg deposits, has found that Hg is in inorganic form only in the surface of the bark, whereas in deeper layers it appears as Hg-cysteine or Hg bound to tannic acid [31]. The authors of that study suggest that, after Hg has been captured at the bark surface as particulate (or as gaseous Hg), a stable chemical bond is established with organic ligands from the substrate, thus resulting in organic compounds or complexes. Unfortunately, cinnabar and metacinnabar, which are almost insoluble and poorly available, are the most common inorganic compounds in a mining environment. It is therefore more likely that Hg^0^ and/or Hg^2+^ are the species involved in this formation process for Hg-cysteine or Hg bound to tannic acid.

In summary, the THg quantified in bark likely corresponds to a combination of two processes: uptake of gaseous Hg via the lenticels to produce Hg-cysteine and Hg bound to tannic acids inside the bark; and uptake of particulate matter with cinnabar and Hg bound to humic acids, which would be retained on the surface of the barks. This results in the identification of only two Hg species upon thermal speciation as the desorption temperatures of Hg-cysteine, Hg bound to tannic acids and Hg bound to humic acids are very similar (210–220 °C), thus being indistinguishable. As a consequence of this, the THg concentrations in bark should be considered to be a biosensor of total atmospheric Hg (TAM), taken to be the sum of Hg^0^, Hg^2+^ and PBM. It has not yet been possible to determine, based on the data of this study, whether the proportions between the three species are maintained in the plant tissue analyzed, or if there are factors that modify this, such as the reversible character of the uptake of Hg^0^ and Hg^2+^, the re-emission of these two species, or the fixing capacity of the particles on the surface of bark, amongst others.

The combined use of the two plant tissues from *Platanus hispanica* would allow an evaluation of the TGM emission sources, including Hg^0^ and Hg^2+^ with the THg content in leaves, and the PBM emission sources by a combined use of the THg content in leaves and bark. Although it is not possible to evaluate the different rates and efficiencies of TGM uptake by these tissues, they are very likely to be different if we take into account that, in leaves, it is plant gas exchange via stomata that drives the uptake, whereas in bark this function does not exist and it is expected that the rate of exchange of water vapor and gases via the lenticels, and therefore also the rate of TGM uptake, will be lower. Nevertheless, assuming that these two absorption rates remain constant for the same type of plant tissue in the entire study area, it may be possible to obtain a Particulate-bound Mercury Factor (PBMF) from the THg_leaves_/Thg_bark_ ratio. This PBMF ratio will allow us to delimit the main PBM emission sources in the area studied. The distribution map thus obtained (Figure 5) clearly shows that the main source of PBM emission is in the soils located to the south of the urban area of Almadén, both in the highly contaminated soils of the orchards area to the south of the metallurgical enclosure (zone 3 in the Figure 1), as well as those of the band of pine forest, which have THg contents that decrease from 280 mg kg^−1^ to the west, to 28 mg kg^−1^ to the east, next to the contaminated roads [24]. Thus, the combined use of the THg content of leaves and bark results in two distribution maps that perfectly delimit the TGM (Figure 2B) and PBM (Figure 5) emission sources.

## 5. Conclusions

The feasibility of two plant tissues from a tree commonly found in urban environments has been evaluated for the biomonitoring of gaseous Hg species in a mining setting. It has been possible to determine that the processes for Hg uptake from the atmosphere are different in each tissue, with leaves bioaccumulating only gaseous Hg (Hg^0^ and Hg^2+^), preferably in daylight hours, and bark absorbing a combination of TGM and PBM both during the day and at night. Thus, indicative maps for the main sources of TGM and PBM emissions have been obtained, thereby perfectly delimiting the main TGM and PBM sources in the urban area of Almadén.

This method complements TGM biomonitoring systems already tested with other urban trees by adding the detection of PBM emission sources and, therefore, biomonitoring all Hg species present in the atmosphere.

Although its application has been restricted in the present work to a mining environment, with inorganic Hg compounds in soils and sediments and Hg^0^ emissions, other scenarios should be evaluated to determine the scope of this method for the biospeciation of Hg in the atmosphere with other atmospheric Hg fractionation, for example by application to emissions from chloralkaline plants with a high Hg^2+^ content.

## Figures and Tables

**Figure 1 ijerph-18-05191-f001:**
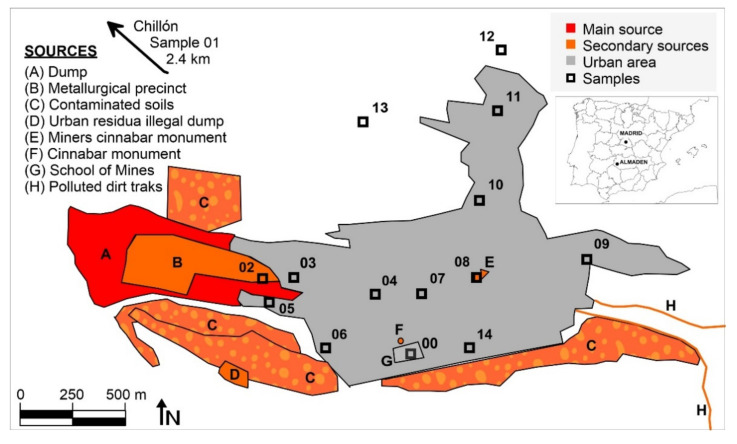
Location of sampling sites and main Hg emission sources in the area studied. Sample codes can be found in Table 1.

**Figure 2 ijerph-18-05191-f002:**
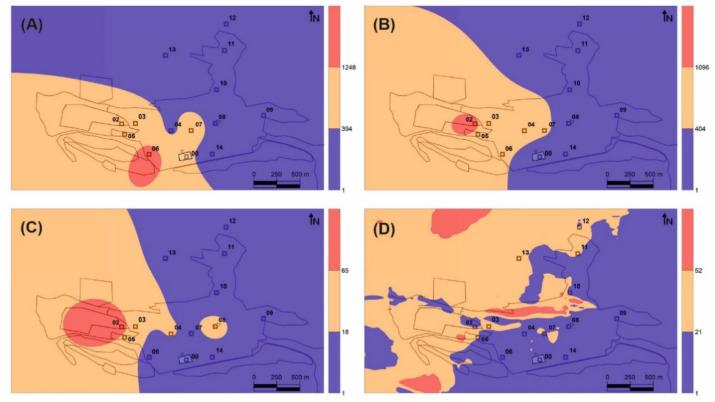
Distribution maps of THg content in bark (**A**) and leaves (**B**) and TGM in Almadén based on study sites (**C**) and on portable measurements using 6917 sampling locations covering the entire urban area (**D**). THg contents are expressed in ng g^−1^, while TGM levels are expressed in ng m^−3^.

**Figure 3 ijerph-18-05191-f003:**
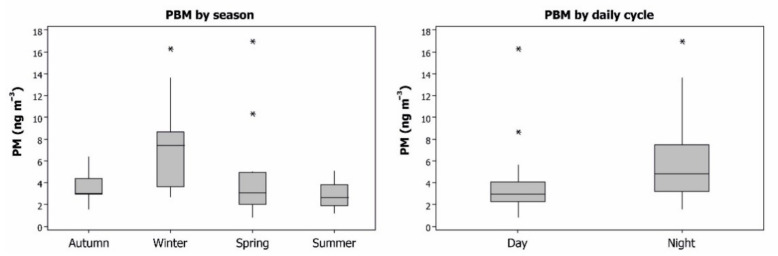
Boxplots of PBM concentrations in the Almadén area by season and by day/night cycle. Outliers appear with the symbol (*).

**Figure 4 ijerph-18-05191-f004:**
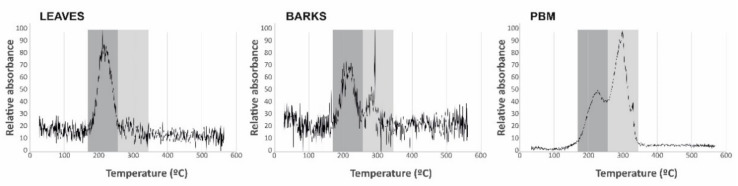
Thermal desorption profiles for leaf, bark, and PBM samples. The main species identified are highlighted in the shaded areas.

**Figure 5 ijerph-18-05191-f005:**
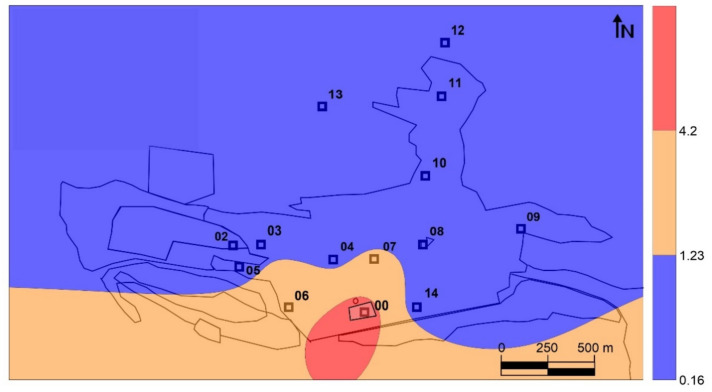
Distribution map of the PBMF obtained as a ratio between THg_leaves_/THg_bark_.

**Table 2 ijerph-18-05191-t002:** A single Pearson correlation matrix between the main factors studied: THg in leaves and barks, and TGM concentrations in the 14 sites, in a complete survey and during the night hours.

	Barks THg	Leaves THg	TGM Sites	TGM Survey
Leaves	0.422			
	*0.118*			
TGM sites	0.189	**0.890**		
	*0.499*	*0.000*		
TGM survey	**0.782**	**0.701**	0.462	
	*0.001*	*0.004*	*0.083*	
TGM night	−0.130	−0.064	−0.091	−0.112
	*0.645*	*0.820*	*0.748*	*0.691*

Significative correlations appear in bold while the degree of significance (ρ) appears in italics.

## Data Availability

The raw data is available on request to the Instituto de Geología Aplicada (UCLM) at secreteria.igea@uclm.es.

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
