# Peer review of "Biomonitoring of Hg0, Hg2 and Particulate Hg in a Mining Context Using Tree Barks+"

_ijerph, 2021, doi:10.3390/ijerph18105191_

Round 1

Reviewer 1 Report

The manuscript titled "Biomonitoring of total atmospheric mercury (Hg0, Hg2+ and particulate Hg) in a mining context using tree barks " aims to explore the potential use of two plant tissues from a tree to assess the mercury level found in the air. This topic is pertinent given the need to develop environmental tools to follow mercury contamination in the atmosphere as well as to better identify the Hg contamination sources. While I do find this topic worthy of study, there are major improvements that should be made before this is accepted for publications.

General comments:

A big issue I have with this study is the low number of sites sampled (n = 14) and of samples collected from these sites (bar, leaves). In addition to this limitation, the statistical treatment of data was hardly performed, leading to potential flaws in the result interpretations and discussion.

For examples:

-Table 1 shows the mean and the standard deviation (SD) of THg measurements in bark and leaves, but never the number of replicates used in such analysis. Why not apply statistical tests (ANOVA-one way, Kruskal-Wallis test on the rank) to distinguish differences among sample sites (from sites 0 to 14).

To validate the feasibility of two plant tissues for biomonitoring purposes, you must explore if Hg measurements (GEM, PBM or Total Hg) in the atmosphere and Hg bioaccumulated in both tissues (bark and leaves) are significantly correlated. These correlations, which are critical to accomplish the main objective of this study, was only mentioned in the text (on page 7, lines 268-269) but not presented in Results sections.

Figs. 2 and 5:  Do you think that 14 points are sufficient to generate spatial differences as presented in these figures? What is your spatial resolution here?  How do you establish your levels to distinguish one area from the other?

Figure 3: You presented some mean values with the standard errors but noting is mentioned about the replicate number (n). Again, why not to apply statistical tests (ANOVA-one way, Kruskal-Wallis test on the rank) to distinguish differences among seasons?

The authors prepared composite samples for leaves and bark in tree samples, but what is the constitution of these samples, I mean how many leaves you collected for each composite sample? Similar question concerning the tree barks.

Another point to make clear is the way in which the author measured the variables described in the document concerning Hg contamination in the atmosphere: GEM, RGM, TGM, and PBM.  More explanation should be addressed to better understanding how the authors distinguish one form from the other.

Specific comments:

Page 1

-Lines 6-11, Please be consistent in the way Cuidad real, Spain is written the text.

-Line 17: You wrote, “between atmospheric Hg compound”. You should write between atmospheric Hg phases (gaseous,particulate).

Introduction

Some phrase stated by the authors required the inclusion of the pertinent references to support the idea:

-Line 54: “the possibility of determining it directly using gaseous samplers” (Reference).

-Line 58: total gaseous Hg (TGM), which is the sum of GEM and RGM (Reference).

-Line 64; periods of time in a simple and inexpensive manner (Reference)

-Line 67. Organic tissues, it exists inorganic tissues?

In this section, I find more ideas concerning the use of organisms (lichen, moose, and tree) as biomonitor for Hg contamination should be more developed. There are a lot of text about Hg distribution among phase (particulate, gaseous, etc.)

Line 81. If you objective is “to evaluate tree barks as a biomonitor for PBM in a mining environment” why you mentioned the leaves of Platanus hispanica ? Please re-write your idea in these sentences.

Materials and Methods

Page 3

Lines 112-124:

Do you have a criterion of the tree barks you collected (in terms of deep, levels of deterioration) in each sampled site?

In your preparation of composite samples, what is the tree age you included in the samples?

How many leaves and tree barks were used to prepare your composite samples?

In the collection of these samples, did you consider the wind direction in your sites?

Do you think deionized water is appropriate to removed particles in the surface of your leaves and barks?

Any treatment in your samples, bottles, material used in the sampling collection to avoid Hg contamination?

Fig.1 I strongly recommend adding a map of whole Spain indicating the location of .

Page 4

-Lines 133-134.

This is the only place the authors indicted the number of replicates (n = 3). Please, provide this information in the figures prepared in this study.

Please, provide the recovery (in %) of the hg measured in the SRM 1515 (Apple leaf). Did you perform any blank in your measurements?

Line 135: Are you sure you are determining Hg species?

Lines 171-174: You mentioned in the text all these variables. However any discussion concerning these variables was found in the text.

Results

Page 5

Line 182: Why not to apply statistical tests (ANOVA-one way, Kruskal-Wallis test on the rank) to distinguish differences among sample sites (from site 0 to 14). Are the differences mentioned statistically significant?

Page 6

Lines 206-209. I did not see the relationship mentioned between leaf Hg and TGM levels, please add a graphic including such relationships (see line X).

I have a strong concern about the resolution of the figure 2A, 2B, 2C and 2B with only 14 sites.

Any information about the wind direction in these sampled sites?

Line 224: the PBM data gave as an average included the measurements obtained in all sampled sites? If yes, why to do that? There are not differences among sampled sites?

Page 7

Lines 239:  You presented some mean values with the standard errors but noting is mentioned about the replicate number (n). Again, why not to apply statistical tests (ANOVA-one way, Kruskal-Wallis test on the rank) to distinguish differences among seasons?

Line 242: I am not sure if Hg speciation is the correct term here. Why not use elemental spectrometry or molecular spectrometry to better identify the Hg speciation estimated?

Lines 248-254: Please, how you do the % calculation mentioned? Why not compare the area peak observed for each standard and those measured in each site? It could be interesting if the % found in bark and leaves varies as a function of Hg contamination. It is important to show not only qualitative results.

Lines 260-263. Please rewrite the sentence written in these lines.

Page 8

Lines 272-292:

The way the authors proposed to identify emission sources of Hg in the area studied would require confirmation studies using other analytical approaches: isotopic signatures of Hg from the difference sources and then to compare them with those observed in leaves and back of trees. It could be very helpful to draw any idea about these perspectives.

Limes 298-310.

The lack of correlation should also be attributed to a low number of sampled sites used in this study.

I am not sure that fig. 2 is suitable to explain the difference observed between leaves and barks.

Page 10.

See comment on figure 5 in the first section.

Reviewer 2 Report

The paper is very interesting but it is necessary to review some concepts, methodology, results and conclusions. My suggestions and doubts are in the pdf file.

Reviewer 3 Report

The paper examines mercury (Hg) exposure in the area of Almadén, a long-standing mining town. The authors measured gaseous metallic Hg in air, total mercury in the bark and leaves of plane trees (at 14 different sites), and particle bound Hg at one site. They also performed a speciation analysis of Hg in leaves, barks and particles.

The authors report a strong spatial correlation between gaseous Hg and Hg in leaves. In leaves all the Hg is in the form of organic compounds. The Hg content in the barks was not spatially correlated with that in leaves or with gaseous Hg in air samples. As with particle bound Hg the Hg in barks consisted of Hg in organic compounds and of inorganic species (cinnabar).

The authors conclude from this findings, that leaves directly (through their stomata) take up the gaseous metallic Hg, while barks take up Hg from the particle bound fraction. This latter conclusion is not directly supported by data because they do not have spatially diverse data on particle bound Hg concentrations. They base their conclusion on the facts, that Hg in barks is also found in inorganic form, and that the spatial distribution in barks differs from the distribution of the gaseous Hg. They also state that uptake of Hg through the roots directly from the ground is not possible. This assumption is backed by two references [17, 18], which I am not going to read in detail. I am not familiar with the uptake behaviour of plane trees. But I do know for some plants that uptake from the soil also depends on soil chemistry and speciation of Hg. So maybe the claim that uptake also into barks must occur directly from the air, is a bit rash?

I feel the authors should also discuss temporal variation in exposure. While leaves only persist for one year and Hg concentration in leaves therefore reflects the exposure of the last year only, bark persists longer and therefore Hg in the bark would be affected also by exposure in previous years.

We learn that the city of Almadén has experienced a great change in very recent years with the closure of the mine and metallurgical production sites and with the sanitation of waste dumps. This might well have introduced a change in the spatial distribution of gaseous atmospheric Hg. Analysing different parts of the tree (leaves and bark) would therefore not only be a means to analyse exposure to different Hg species, but would maybe also allow to study the temporal course or change in exposure. Of course, this would need further research also.

Nevertheless, the authors should also discuss this possibility in their paper!

Round 2

Reviewer 1 Report

The authors have included significant changes in the new version of the manuscript. They also explained some limitations related to the methodology and experimental design used in their work. Clearly, the last version is worthy of being published.